# Federated Learning with Unlabeled Clients:
# Personalization Can Happen in Low Dimensions

**Hossein Zakerinia** [* 1]   **Jonathan Scott** [* 1]   **Christoph H. Lampert** [1]

## Abstract

Personalized federated learning has emerged as a popular approach to training on devices holding statistically heterogeneous data, known as clients. However, most existing approaches require a client to have labeled data for training or finetuning in order to obtain their own personalized model. In this paper we address this by proposing FLowDUP, a novel method that is able to generate a personalized model using only a forward pass with unlabeled data. The generated model parameters reside in a low-dimensional subspace, enabling efficient communication and computation. FLowDUP's learning objective is theoretically motivated by our new transductive multi-task PAC-Bayesian generalization bound, that provides performance guarantees for unlabeled clients. The objective is structured in such a way that it allows both clients with labeled data and clients with only unlabeled data to contribute to the training process. To supplement our theoretical results we carry out a thorough experimental evaluation of FLowDUP, demonstrating strong empirical performance on a range of datasets with differing sorts of statistically heterogeneous clients. Through numerous ablation studies, we test the efficacy of the individual components of the method.

## 1. Introduction

Federated learning (FL) (McMahan et al., 2017) is a widely adopted approach to enable privacy-preserving machine learning in distributed environments. Data-holding devices (clients) cooperatively train predictive models under the coordination of a central server, without sharing their local data. Differing underlying behaviors often result in client data following different distributions, potentially rendering single global model training ineffective (Kairouz et al., 2021). Personalized federated learning (Smith et al., 2017) has emerged as a popular approach to address this challenge of statistical heterogeneity. It enables clients to learn personalized models while still leveraging collective knowledge from the broader network. However, since most approaches to personalization rely on some form of finetuning on the client's local data they require a client to have labeled data in order to obtain a personalized model. This poses a major challenge for personalization in applications where active participation from the client (i.e. user) is required to create labels for their local data. In such scenarios, even though some clients will likely be *labeled* (have data with labels), the majority of potentially participating clients will likely be *unlabeled* (i.e. have no labels for their data). Similarly, most future clients will potentially also be unlabeled.

In this paper we address this issue with FLowDUP: **F**ederated **Low-D**imensional **U**nlabeled **P**ersonalization. **FLowDUP learns to generate a personalized model for any client, even if the client has only unlabeled data.** To do so, it trains a hypernetwork (Ha et al., 2017) that takes as input an unlabeled dataset and outputs the parameters of a personalized predictive model. Crucially, rather than outputting all the parameters of a personalized model, the hypernetwork instead outputs a parametrization in a subspace of much smaller dimension than the underlying personalized model. The full model is obtained by multiplying the subspace parameters with a fixed (random) *expansion matrix*. The use of a low-dimensional subspace is a crucial contribution of our work. It allows for the generation of large model architectures and means that the hypernetwork is small and efficient enough to be transmitted to and run on the client devices. This allows FLowDUP to adhere strictly to the federated paradigm that clients' data should not be transferred away from the device. Without it, hypernetwork-based methods are only able to generate very small models and are only practical if the hypernetwork runs on the central server (Shamsian et al., 2021; Amosy et al., 2024; Scott et al., 2024), which is at odds with the FL principle that

---

[*]Equal contribution   [1]Institute of Science and Technology Austria (ISTA). Correspondence to: Hossein Zakerinia <hossein.zakerinia@ista.ac.at>, Jonathan Scott <jonathan.scott@ista.ac.at>, Christoph H. Lampert <chl@ista.ac.at>.

*Proceedings of the $43^{rd}$ International Conference on Machine Learning*, Seoul, South Korea. PMLR 306, 2026. Copyright 2026 by the author(s).

clients' data never leaves the clients' device.

We propose a learning objective for FLowDUP that consists of two parts. The first part measures the quality of the generated client models, for which it exploits that some of the clients at training time do have labels. The second part prevents overfitting by penalizing large deviations between the generated models and a learned regularization term. Evaluating it requires only unlabeled data, so it can be computed from all clients available during training. FLowDUP and its learning objective are motivated by theoretical results derived in a multi-task framework. Specifically, we prove a generalization bound that provides performance guarantees for unlabeled clients. Our learning objective is derived by optimizing terms that appear in this bound.

In addition to our theoretical contributions we carry out an experimental evaluation over a range of datasets, exhibiting various types of statistical heterogeneity, and at a range of proportions of labeled clients present during training. We conduct additional experiments and ablation studies to better understand the different components of FLowDUP.

To summarize, our contributions are as follows:

- We propose FLowDUP, a method that generates low-dimensional personalized model parameters using only an on-device forward pass on unlabeled data.

- We derive generalization bounds in a multi-task framework, which, in our FL setting, provide guarantees on FLowDUP's performance on unlabeled clients.

- Based on these bounds we propose a theoretically motivated training objective that is able to also leverage unlabeled clients during training.

- We conduct an experimental evaluation demonstrating strong empirical performance of FLowDUP and illustrate the efficacy of the individual components with ablation studies.

## 2. Background

**Notation**    We assume a federated setting with $n$ clients participating in training. Each client $i \in \{1, \ldots, n\}$ possesses a data distribution $D_i$ over shared inputs and output sets, $\mathcal{X} \times \mathcal{Y}$. The clients are statistically heterogeneous, meaning that $D_i \neq D_j$ for $i \neq j$ is possible. We call $n_L$ the number of clients that hold labeled data and $n_U$ the number that hold only unlabeled data, i.e. $n_L + n_U = n$. Writing $S_i$ for the dataset of client $i$, and $m_i = |S_i|$, we then have

$$S_i := \begin{cases} (X_i, Y_i) = \left(x_i^j, y_i^j\right)_{j=1}^{m_i} \sim D_i & \text{if } i \in \mathcal{I}_L, \\ X_i = \left(x_i^j\right)_{j=1}^{m_i} \sim D_{i|\mathcal{X}} & \text{if } i \in \mathcal{I}_U, \end{cases} \quad (1)$$

where $D_{i|\mathcal{X}}$ denotes the marginal distribution of $D_i$ over $\mathcal{X}$. We denote by $f(\cdot\,; \theta) : \mathcal{X} \to \mathcal{Y}$ a predictive model

parameterized by $\theta \in \mathbb{R}^d$, and by $\ell : \mathcal{Y} \times \mathcal{Y} \to \mathbb{R}$ a loss function, such that $\ell(y, f(x; \theta))$ measures the prediction quality of $f(\cdot\,; \theta)$.

**Learning in a subspace**    Modern neural networks use models with many parameters, often more than the number of training examples. However, it has been shown that their *intrinsic dimensionality* is much smaller than their number of parameters (Li et al., 2018), i.e. it is possible to learn strong models in a random subspace of much smaller dimension than the full dimension of the model parameter space. Formally, to learn model parameters $\theta \in \mathbb{R}^d$, given an initialization model $\theta_0 \in \mathbb{R}^d$, and a random expansion matrix $P \in \mathbb{R}^{d \times k}$ (which describes the basis of a random subspace), we can describe a model in the generated random subspace by learning a vector $v \in \mathbb{R}^k$ as

$$\theta = \theta_0 + Pv. \quad (2)$$

Prior work in standard learning (Li et al., 2018) and recently multi-task learning (Zakerinia et al., 2025) showed that $k$ can typically be chosen orders of magnitude smaller than $d$ while still allowing high accuracy models to be trained.

In this work, we keep the matrix $P$ and the initialization $\theta_0$ fixed, such that $\theta$ is completely determined by $v$. With a slight abuse of notation, we then also write $f(\cdot\,; v)$ as shorthand for $f(\cdot\,; \theta_0 + Pv)$. We use the random expansion introduced in (Lotfi et al., 2022), which constructs the matrix via the Kronecker product of two smaller matrices to enable efficient computation. See Appendix B.2 for more details.

## 3. Personalized Federated Learning with Unlabeled Clients

### 3.1. Training Objective

Our goal is to generate personalized models for clients that possess only unlabeled data. We assume we have access to labeled and potentially also unlabeled clients during training, and that future clients we will encounter might be unlabeled. The primary object we work with is a hypernetwork $h : \mathcal{P}(\mathcal{X}) \to \mathbb{R}^k$, where $\mathcal{P}(\mathcal{X})$ denotes the power set of $\mathcal{X}$. The goal of $h$ is to take in an unlabeled client dataset and output the (low-dimensional subspace) parameters of a personalized model that works for the underlying client data distribution. Formally, for (client) data $X_i \subset \mathcal{X}$, $h$ is defined as

$$h(X_i) := h_2 \left( \frac{1}{|X_i|} \sum_{x \in X_i} h_1(x) \right), \quad (3)$$

where $h_1 : \mathcal{X} \to \mathbb{R}^e$ is feature extraction module, which is followed by an average across the (batch of) features, and a fully-connected module $h_2 : \mathbb{R}^e \to \mathbb{R}^k$.

To generate model parameters for client $i$, given $X_i$, we first compute $v_i = h(X_i)$, then the full-dimensional parameters

**Algorithm 1** FLowDUP

1: **Input:** Client datasets $\{S_i\}_{i=1}^n$, number of rounds $T$, global learning rate $\eta_g$, labeled client sampling rate $\alpha$
2: Initialize learnable parameters: $\psi$
3: **for** round $t = 1$ to $T$ **do**
4:     Server selects a subset of clients $\mathcal{C}$, with fraction $\alpha$ labeled
5:     **for** each client $i \in \mathcal{C}$ **in parallel do**
6:         Client $i$ receives current parameters $\psi$
7:         Client $i$ performs local updates: $\Delta\psi_i \leftarrow$ `ClientUpdate`$(S_i, \psi)$
8:         Client $i$ sends update $\Delta\psi_i$ to server
9:     **end for**
10:    Server aggregates updates: $\Delta\psi \leftarrow \frac{1}{|\mathcal{C}|}\sum_{i\in\mathcal{C}} \Delta\psi_i$
11:    Server updates parameters:
       $\psi \leftarrow$ `GradientUpdate`$(\psi, \Delta\psi\,;\, \eta_g)$
12: **end for**
13: **Return:** Final parameters $\psi$

---

for $f$ are obtained by random expansion: $\theta_i = \theta_0 + Pv_i$ as defined in (2). The final personalized model is $f(\cdot\,;\,\theta_i) : \mathcal{X} \to \mathcal{Y}$. Note that $P$ is a random matrix and the initialization $\theta_0$ is typically random, these are fixed before training, and can be generated on the client using an agreed-on random seed, and do not need to be transmitted via the network.

We denote the trainable parameters of $h$ by $\psi_h$. Additionally, FLowDUP training also uses a learnable regularization, $\psi_r \in \mathbb{R}^k$, in the low-dimensional model subspace to prevent overfitting. The learnable parameters of FLowDUP are therefore $\psi := (\psi_h, \psi_r)$.

Training FLowDUP means to learn parameters $\psi$ that, given unlabeled data $X$, are able to output personalized client model parameters that work on the distribution that $X$ was drawn from. Our proposed training objective for doing this has two parts, a loss $\mathcal{L}$, and a (learnable) regularizer, $\Omega$.

$$\min_{\psi} \; \mathcal{L}(\psi) + \lambda\Omega(\psi), \qquad (4)$$

The loss measures the quality of a model that is generated for a client, and can therefore be evaluated only on clients that have at least some labeled data,

$$\mathcal{L}(\psi) := \sum_{i\in\mathcal{C}} \sum_{(x',y')\in(X_i',Y_i')} \ell(y', f(x'; h(X_i\,;\,\psi_h))), \quad (5)$$

where $\mathcal{C}$ is a cohort (subset) of clients, $X_i$ is a batch of data without labels from client $i$ and $(X_i', Y_i')$ is a batch of data with labels from client $i$. In case $Y_i'$ is empty, for example because client $i$ has no labeled data, the value of the sum is simply taken to be 0. Intuitively, $\mathcal{L}$ penalizes the hypernetwork parameters $\psi_h$ for outputting models that perform poorly on the clients' data distributions. Notice that

different data batches are used to generate the client model and evaluate it. This is because we want $f$ to be good on the client distribution, and not just on the actual batch used to generate $f$.

The regularizer ensures that the learned models do not diverge too much from each other. It prevents overfitting to individual clients by penalizing large deviations between client model parameters and a global learned regularizer. It can be computed on all clients, as it depends only on unlabeled data:

$$\Omega(\psi) := \sum_{i\in\mathcal{C}} \|h(X_i\,;\,\psi_h) - \psi_r\|^2, \qquad (6)$$

where, again, $\mathcal{C}$ is a cohort of clients and $X_i$ is an unlabeled batch of data from client $i$. Note that, as required for federated learning, the participating clients and data batches can be different in every update step. The regularization strength, $\lambda \in \mathbb{R}$, is a hyperparameter. We provide a more in-depth and formal motivation for our loss function $\mathcal{L}$, based on our generalization bound, in Section 4.

### 3.2. Training Procedure

FLowDUP is designed to be trainable by standard federated learning frameworks. Algorithms 1 and 2 show the necessary steps. Algorithm 1 shows the server steps of FLowDUP that follow a standard Federated Averaging (McMahan et al., 2017) pattern. Training proceeds over a number of rounds. Each round, the server samples a subset of clients with the restriction that a certain fraction, $\alpha$, are clients with labeled data. Each client in the subset receives the current parameters $\psi$ from the server, computes an update to $\psi$ and shares this update with the server. The server then aggregates the client updates and uses the aggregate to update the learnable parameters using an update rule of their choice.

The most important step, namely the `ClientUpdate`, is shown in Algorithm 2. The client receives the latest iteration of the parameters $\psi$ from the server and computes an update direction to those parameters by training on their local data for $E$ local epochs. At each epoch, the client samples a batch of data (line 5) and randomly splits the batch into two equal parts (lines 6, 7). The first part is used to generate the subspace parameter $v$ (line 8), note that no labels are required for this step. The client then computes its contribution to the regularizer value $\Omega$ (line 9). Next, the labeled clients compute the labeled portion of the loss $\mathcal{L}$ (line 11) of the generated client model on the second half of the batch. The total loss is computed (line 12) and the client updates the learnable parameters $\psi$ with a single gradient step (line 13). After all local epochs have been run, the client computes the difference between the final and initial parameters (line 16) and returns this update to the server.

**Algorithm 2** `ClientUpdate`

---

1: **Input:** client $i$, dataset $S_i$, learnable parameters $\psi$, number of local epochs $E$, local batch size $B$, local learning rate $\eta_l$, regularization strength $\lambda$
2: $\psi_{\text{start}} \leftarrow \psi$
3: **for** local epoch $j = 1$ to $E$ **do**
4:     $\mathcal{B} \leftarrow$ client splits $S_i$ into batches of size $B$
5:     **for** each batch in $\mathcal{B}$ **do**
6:         $X_i \leftarrow$ random half of batch $\mathcal{B}$ of data (no labels required)
7:         $(X_i', Y_i') \leftarrow$ remaining half of batch $\mathcal{B}$ data (with labels if available, else $Y_i' \leftarrow \emptyset$)
8:         $v_i \leftarrow h(X_i; \psi_h)$
9:         $\Omega \leftarrow \|v_i - \psi_r\|^2$
10:        $\theta_i \leftarrow \theta_0 + Pv_i$
11:        $\mathcal{L} \leftarrow \begin{cases} \sum_{(x', y') \in (X_i', Y_i')} \ell(y', f(x'; \theta_i)), & \text{if } Y_i' \neq \emptyset \\ 0, & \text{otherwise.} \end{cases}$
12:        $g \leftarrow \nabla_\psi (\mathcal{L} + \lambda\Omega)$
13:        $\psi \leftarrow \texttt{GradientUpdate}(\psi, g; \eta_l)$
        // any suitable optimizers, e.g. SGD or Adam
14:     **end for**
15: **end for**
16: **Return:** parameter update: $\psi - \psi_{\text{start}}$

---

Note that the optimization procedure is an instance of standard federated averaging, and the difference lies only in the objective function. Consequently, under the standard assumptions used in analyses of FedAvg, existing convergence arguments can be applied to the FLowDUP training procedure.

### 3.3. Model Generation

After training, inference with FLowDUP is simple and lightweight, in particular it requires no model training or finetuning, just a single forward pass through $h$. Given a client $i$, with unlabeled data $X_i = (x_i^j)_{j=1}^m$, the client gets $\psi$ from the server and generates a personalized model by passing $X_i$ through the hypernetwork: $v_i = h(X_i; \psi_h)$ and expanding to full dimensionality: $\theta_i = \theta_0 + Pv_i$. The client then has a personalized classifier $f(\cdot; \theta_i)$ that it can use.

## 4. Theory

In this section, we provide a generalization bound for *transductive multi-task learning* in the PAC-Bayesian framework, which provides a theoretical justification for our algorithm. In contrast to *inductive learning*, which uses a labeled training dataset to learn a model for inference on future samples, *transductive learning* (Vapnik, 1998), involves access to a set of unlabeled data for which predictions are desired. Similarly, in transductive multi-task learning, there are both

labeled and unlabeled tasks, and the goal is to learn models for all tasks jointly. As our main theoretical contribution, we prove a PAC-Bayesian generalization bound for transductive multi-task learning for stochastic algorithms that generate a model given an unlabeled dataset. Our proof and results differ from existing works based on PAC-Bayesian bounds for meta-learning (Zakerinia et al., 2024) on the levels at which generalization happens. Prior work can only use the labeled training clients, and no other tasks can participate in the training and guarantee. In our proof, we use different steps and objectives which allow us to also consider the unlabeled clients during the training for regularization and these clients also improve the rates of the bounds and guarantees. Importantly, our objective functions, the learned regularizer, and the role of unlabeled clients in the training directly come from our generalization bounds and motivate different parts of our algorithms theoretically. The proof and the general results are provided in Appendix A. Here we provide a result tailored for our algorithm. Specifically, define a Gaussian distribution over the parameters of the hypernetwork, $\rho_h = \mathcal{N}(\psi_h; \alpha_h \mathrm{Id})$ for a fixed $\alpha_h$, and $n$ posterior models for $n$ clients as $Q_i = \mathcal{N}(h(S_i; \psi_h); \alpha_\theta \mathrm{Id})$, and a regularization distribution as $\mathcal{Q} = \mathcal{N}(\psi_r; \alpha_r \mathrm{Id})$. Our result bounds the gap between the true risk, $\mathcal{R}$, of all clients

$$\mathcal{R}(\rho_h) = \mathop{\mathbb{E}}_{\psi_h' \sim \rho_h} \frac{1}{n} \sum_{i=1}^n \mathop{\mathbb{E}}_{(x', y') \sim \mathcal{D}_i} \ell\big(y', f(x'; h(S_i; \psi_h'))\big),$$

and the training risk, $\widehat{\mathcal{R}}$, of the labeled clients

$$\widehat{\mathcal{R}}(\rho_h) = \mathbb{E}_{\psi_h' \sim \rho_h} \frac{1}{n_L} \sum_{i=1}^{n_L} \frac{1}{m} \sum_{j=1}^m \ell\big(y_i^j, f(x_i^j; h(S_i; \psi_h'))\big).$$

**Theorem 4.1.** *For all $\delta > 0$, and any loss function $\ell : \mathcal{Y} \times \mathcal{Y} \to [0, 1]$, the following statement holds with probability at least $1 - \delta$ over the sampling of $n_L$ clients of $n$ clients and randomness of the dataset. For all parameter vectors, $\psi = (\psi_h, \psi_r)$:*

$$\mathcal{R}(\rho_h) \leq \widehat{\mathcal{R}}(\rho_h) + \sqrt{\left(1 - \frac{n_L}{n}\right) \frac{\frac{1}{2\alpha_h}\|\psi_h\|^2 + c_1}{2n_L}} + \quad (7)$$

$$\mathop{\mathbb{E}}_{\psi_h' \sim \rho_h} \sqrt{\frac{\frac{1}{2\alpha_\theta} \sum_{i=1}^n \mathbb{E}_{\psi_r'} c_i(h, \psi_h', \psi_r') + \frac{1}{2\alpha_r}\|\psi_r\|^2 + c_2}{2mn}}$$

*where $c_i(h, \psi_h', \psi_r') = \|h(S_i; \psi_h') - \psi_r'\|^2$, and $c_1$ and $c_2$ are logarithmic terms in $n$ and $n_L$.*

**Discussion.** Theorem 4.1 provides a direct mathematical justification for FLowDUP. Our overall goal is to achieve high accuracy across all tasks, i.e. minimize the left-hand side of (7). That is not a computable quantity, though, so as a proxy we instead minimize its upper bound on the right-hand side of (7). That consists of $\widehat{\mathcal{R}}(\rho_h)$, which corresponds

*Table 1.* Class-partitioned CIFAR10 for varying fractions, $p$, of the training clients having labeled data. Accuracy on unlabeled test clients.

| Model | CNN | | | ResNet18 | | |
|---|---|---|---|---|---|---|
| $p$ | 0.1 | 0.2 | 1.0 | 0.1 | 0.2 | 1.0 |
| FedAvg | $47.9 \pm 0.1$ | $53.5 \pm 0.4$ | $63.6 \pm 0.4$ | $63.6 \pm 0.9$ | $68.9 \pm 0.7$ | $75.9 \pm 0.3$ |
| FedProx | $47.6 \pm 0.3$ | $53.3 \pm 0.3$ | $63.7 \pm 0.6$ | $63.5 \pm 0.8$ | $68.8 \pm 0.5$ | $75.9 \pm 0.3$ |
| LD-FedAvg | $41.6 \pm 0.6$ | $47.8 \pm 1.1$ | $56.2 \pm 0.7$ | $54.6 \pm 0.8$ | $60.0 \pm 0.5$ | $65.4 \pm 0.1$ |
| FedTTA | $57.2 \pm 1.0$ | $60.2 \pm 1.1$ | $69.7 \pm 3.8$ | $69.3 \pm 2.3$ | $77.2 \pm 1.5$ | $81.9 \pm 2.1$ |
| ATP | $53.5 \pm 2.1$ | $67.3 \pm 4.3$ | $77.8 \pm 2.0$ | $71.3 \pm 2.7$ | $84.6 \pm 1.4$ | $89.6 \pm 0.5$ |
| FLowDUP | $66.1 \pm 1.5$ | $75.3 \pm 1.8$ | $86.6 \pm 0.1$ | $72.7 \pm 1.3$ | $82.9 \pm 1.8$ | $92.3 \pm 0.4$ |

to the training loss across labeled tasks, i.e. FLowDUP's loss $\mathcal{L}$, and some complexity terms. Besides $\|\psi_h\|$ and $\|\psi_r\|$, which are automatically minimized when using optimization steps with weight decay, the latter in particular contains $\sum_{i=1}^{n} \mathbb{E}_{\psi'_r} \|h(S_i \,;\, \psi'_h) - \psi'_r\|^2$, which corresponds to FLowDUP's regularizer $\Omega$. The difference between the theoretical viewpoint and the practical algorithm is that in practice, as it is common in deep learning, we use deterministic neural networks instead of stochastic models, so no expected value operations are required.

On a technical level, Theorem 4.1 has a number of desirable properties. Firstly, it directly reflects the transductive setting, as it controls the risk across all clients in terms of the training loss of just the labeled ones. The complexity term, however, is computed from all clients and its denominator scales with the total number of available samples, which justifies the use of unlabeled clients during training, which contribute to the learning of a shared regularizer. Finally, the sample complexity of the first complexity term is improved by $\sqrt{1 - n_L/n}$ compared to standard inductive bounds, which holds the promise of better between-client generalization in transductive compared to inductive learning.

## 5. Experiments

Here we present our empirical results.[1] In Section 5.1 we explain our setup, in Section 5.2 we include our main results and in Section 5.3 we provide additional experiments and ablation studies to better understand FLowDUP. Implementation details and ablation studies can be found in Appendix B. The values in all results tables are given as the mean and standard deviation of runs over 3 random seeds. We implement our experiments using the `pfl-research` library (Granqvist et al., 2024).

---

[1]Our code can be found at https://github.com/hzakerinia/FLowDUP

### 5.1. Experimental Details

**Baselines.** Most personalized federated learning methods do not apply to our setting, because they need to compute gradients using labeled data in order to personalize on the client. Since our goal is generating personalized models for clients with only unlabeled data we compare only to methods that are capable of doing this. These methods, like ours, utilize labeled clients during the training phase. The simplest to consider are those that train a single (full-dimensional) global model $f$ on just the labeled clients without any personalization. This of course results in a predictive model that can be used by any unlabeled client. In this category we include Federated Averaging (FedAvg) (McMahan et al., 2017) and FedProx (Li et al., 2020). We also include a baseline that we call LD-FedAvg that trains the low-dimensional subspace parameterization of $f$, using federated averaging. This baseline therefore has the same client model $f$ parameterization as the personalized models generated by FLowDUP. Finally, we include the two federated test-time adaptation methods that are applicable and can personalize without labeled data: FedTTA (Ye et al., 2024) and ATP (Bao et al., 2023). Both aim to personalize a globally trained model $f$, at prediction time, to an unlabeled client with gradient updates obtained from an unsupervised loss. Even though the goal of the paper is to generate models for unlabeled clients, in addition to the main experiments, we also compare our method to baselines that have access to labeled data in Section 5.3.

**Datasets** We include a range of datasets with different types of statistically heterogeneous clients. Following prior work in personalized FL we simulate label heterogeneity with a non-i.i.d. partitioning of CIFAR-10 (Krizhevsky, 2009), where each client receives 100 data points from only 2 classes. We simulate heterogeneity in the features $\mathcal{X}$ by partitioning and rotating Fashion-MNIST (Xiao et al., 2017) and MNIST (LeCun et al., 1998), as first done by Ghosh et al. (2020). Specifically, we do this by creating an i.i.d. partitioning of the dataset into clients, where each client receives 100 randomly i.i.d. sampled data points. Then each client randomly samples a rotation from $\{0°, 90°, 180°, 270°\}$ and

*Table 2.* Rotated Fashion-MNIST (left) and FEMNIST (right) for varying fractions, $p$, of the training clients having labeled data. Accuracy on unlabeled test clients.

| Dataset | Rotated Fashion-MNIST | | | FEMNIST | | |
|---|---|---|---|---|---|---|
| $p$ | 0.1 | 0.2 | 1.0 | 0.1 | 0.2 | 1.0 |
| FedAvg | $77.7 \pm 0.1$ | $79.2 \pm 0.2$ | $81.4 \pm 0.6$ | $76.5 \pm 0.3$ | $78.0 \pm 0.3$ | $84.3 \pm 0.1$ |
| FedProx | $77.3 \pm 0.7$ | $78.7 \pm 0.4$ | $80.0 \pm 0.6$ | $72.3 \pm 0.4$ | $74.8 \pm 0.4$ | $83.3 \pm 0.0$ |
| LD-FedAvg | $76.1 \pm 1.1$ | $78.8 \pm 0.3$ | $81.3 \pm 0.1$ | $61.4 \pm 1.4$ | $74.4 \pm 0.3$ | $80.6 \pm 0.4$ |
| FedTTA | $78.3 \pm 0.5$ | $80.2 \pm 0.1$ | $81.2 \pm 0.2$ | $71.1 \pm 2.4$ | $72.6 \pm 3.5$ | $74.3 \pm 3.1$ |
| ATP | $76.7 \pm 0.8$ | $79.7 \pm 0.7$ | $81.5 \pm 0.7$ | $82.9 \pm 0.1$ | $83.7 \pm 0.2$ | $84.1 \pm 0.4$ |
| FLowDUP | $81.5 \pm 0.3$ | $83.3 \pm 0.5$ | $87.3 \pm 0.2$ | $84.5 \pm 0.4$ | $86.4 \pm 0.3$ | $89.3 \pm 0.1$ |

rotates all of their images by that amount. Finally, we also include results for FEMNIST (Caldas et al., 2018), a federated dataset with 62 classes and an existing partition into around 3500 clients, holding in total $\approx 800000$ datapoints. The clients in FEMNIST exhibit both feature (different writing styles) and label (different class frequencies per client) heterogeneity. For each of the datasets we simulate a range of different levels of label prevalence in the clients. Specifically, for $p \in \{0.1, 0.2, 1.0\}$ we randomly choose $pn$ of the $n$ clients to have labels and the rest are fully unlabeled.

**Network architectures** We run experiments using two different architectures for the client model $f$, a CNN following prior work (McMahan et al., 2017) and a ResNet18 (He et al., 2016). On partitioned CIFAR10 we present results for both the CNN and the ResNet18, for the remaining datasets we use a CNN. For FLowDUP recall the structure of $h$ is: $h(X) = h_2(\frac{1}{|X|} \sum_{x \in X} h_1(x))$. For our main results in Section 5.2 we always implement $h_1$ using the same architecture (CNN or ResNet18) as $f$ for that particular setting. In Section 5.3 we explore the case when they differ. We implement $h_2$ as a fully connected network with a single hidden layer. Both FLowDUP and LD-FedAvg work with subspace parameterizations of the client models $f$. For the results in Section 5.2 we set the subspace dimension to $k = 10^4$. This represents an approximately $15\times$ reduction in the number of client model parameters for the CNN and $1100\times$ for ResNet18. We examine the impact the choice of $k$ has in 5.3. For FedTTA the global model architecture is the same as that used for FedAvg and the adaptation model is the same 3-layer network as used in (Ye et al., 2024).

**Comparison of the number of training parameters** If we denote the number of model parameters by $d$ and the number of hidden units before the last layer by $p$, the final layer would have $p \times d$ parameters, whereas for FLowDUP it would be $p \times k$. For example, assume $d = 12M$ (e.g., ResNet-18), p=256, $k = 10^4$. Prior work would need $256 \times 12M \sim 3.07$ billion weights ($\sim 12.3$ GB in fp32) just in the hypernetwork's output layer (As in (Shamsian et al., 2021; Scott et al., 2024)). In contrast, FLowDUP only requires $256 \times 10^4 \sim 2.56$ million weights ($\sim 10$ MB).

### 5.2. Experimental Results

Our main results are shown in Tables 1 and 2. Table 1 shows class-partitioned CIFAR10 as we vary the fraction of labeled clients present in the training set ($p$). The left part of the table shows the results for when the model architecture is a CNN and the right part for ResNet18. The results show that, with the exception of $p = 0.2$ for ResNet18, FLowDUP performs best across the board. As expected all methods improve as the fraction of labeled clients increases and when we switch from using a CNN to a ResNet18. Notably, even when using the CNN architectures, FLowDUP is able to outperform the non-personalized baselines using a ResNet18. We also observe that LD-FedAvg has significantly lower performance than FedAvg. This shows that in fact for a single global predictive model, lower subspace dimension reduces the power of the model. However, FLowDUP's strong performance shows that, given a well learned representation of client heterogeneity in $h$, a low-dimensional subspace model can obtain good personalized performance.

Table 2 shows the results for Rotated Fashion-MNIST (left) and FEMNIST (right). The results for Rotated MNIST can be found in the Appendix in Table 6. Here all methods use the CNN architecture. When comparing to CIFAR10, for both these datasets the test-time adaptation methods, FedTTA and ATP, perform quite poorly. This is to be expected given that they adapt the global model based on the output logits of the unlabeled data. When statistical heterogeneity is present only in the labels (as is the case for class-partitioned CIFAR10), the logits are a good summary of the client personalization requirements. However, in the presence of feature heterogeneity the logits alone provide a poor adaptation signal. FLowDUP performs better as it personalizes based on all the clients' feature data.

### 5.3. Additional Studies

Here we provide results from ablations investigating the components of FLowDUP. In addition to these we include in the appendix a study of the effect that training with unlabeled clients has (Tables 7 and 8) as well as the benefits of

the learnable regularizer $\psi_r$ (Table 9), and finally the effect of the labeled client sampling parameter $\alpha$ (Table 10).

**Differing client model architectures** We train $h$ to output personalized model parameters in a subspace of dimension $k$. However, for any given $k$ we are still free to choose the architecture of the client model $f$. Moreover, the choice of architecture of $f$ does not affect the communication cost of the training procedure (which is critical and typically the primary bottleneck when running federated training in practice). It does, however, affect the compute cost on the client side. On the flip side, the choice of hypernetwork architecture does affect the communication cost as $h$ is transmitted from server to client. This gives FLowDUP some flexibility when trading off communication vs computational cost. In Table 3 we show results for CIFAR10 for different combinations of hypernetwork and client model architectures. The results show that for a fixed subspace dimension $k$, changing the architecture of $h_1$ or $f$ can have a substantial impact on performance. Most notably, when comparing row 1 with 2, and row 3 with 4, we see that changing $f$ from a CNN to a ResNet18 gives a performance boost of 4-5% without incurring any additional communication cost.

**Varying the subspace dimension** While the choice of client model architecture affects only the computational cost of FLowDUP, the subspace dimension $k$ affects both communication and computational cost. This is because the final layer of $h$ has dimension $k$ and the matrices in 2 scale with $k$. Table 4 shows results for FLowDUP on CIFAR10 as we vary $k$. In these experiments we fix the architectures of $h_1$ and $f$ to be CNN. We observe that performance increases monotonically with $k$. Most notably we observe that, with the exception of $p = 0.1$, FLowDUP outperforms the baselines, even for very small $k$, which in turn incur low compute costs on the client. This means $k$ can be chosen based on the highest value supported by the communication costs, and no hyperparameter tuning is needed.

**Understanding dataset embeddings** Recall the hypernetwork architecture from (3). We can interpret the input to $h_2$ as an embedding representation of the client dataset $X$, which we call $r(X) := \frac{1}{|X|} \sum_{x \in X} h_1(x)$.

To better understand the inner workings of this mechanism we visualize the space of dataset embeddings. Concretely, for each validation client in the Rotated Fashion-MNIST dataset, we compute $r(X_i)$ for the $h$ that was trained on the training clients. In our setting, the representations are 256-dimensional, therefore we use dimensionality reduction for visualization. We separately apply PCA and t-SNE (van der Maaten & Hinton, 2008) to project the representations into 2 dimensions. The plots (Figure 1) show that FLowDUP learns to organize the space of client dataset representations according to the present statistical heterogeneity, namely that the representations are clustered according to which

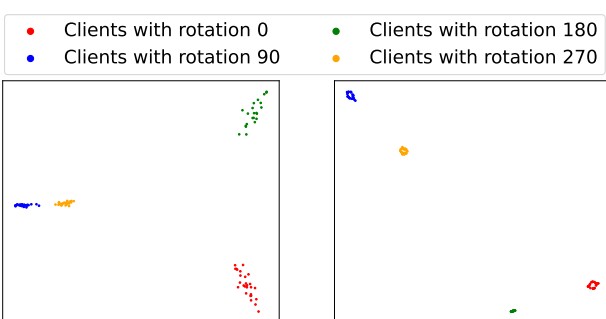

*Figure 1.* Visualization of the client embeddings on Rotated Fashion-MNIST. Projection to two dimensions using PCA (left) and t-SNE (right).

rotation the data has. Note that nowhere in the learning objective is this hard-coded. Rather FLowDUP has learned the type of heterogeneity present in the client datasets, and organized the representation space, roughly speaking, clients with the same rotation receive the same model.

**Comparison with baselines with labeled data:** We also report comparison when clients have access to labeled data. We use the baselines that require labeled data and are unable to generate a model in the main setting of our paper, i.e., personalization with unlabeled data. We report the results from (Scott et al., 2024), for Per-FedAvg (Fallah et al., 2020), FedRep (Collins et al., 2021), pFedMe (Dinh et al., 2020), kNN-Per (Marfoq et al., 2022), pFedHN (Shamsian et al., 2021), and PeFLL (Scott et al., 2024).

The setup of this experiment is the same as our results for CNN in Table 1 and FEMNIST in Table 2, when $p = 1$. The results for FLowDUP use labeled clients to train the hypernetwork, however does not use labels to generate the final personalized model. As shown in Table 5, our method outperforms all the baselines except PeFLL, which is slightly better than FLowDUP. However, this performance comes at the cost of higher communication overhead and access to individual client updates, as discussed in Section 6.

# 6. Related Work

**Personalized federated learning for unlabeled clients** First proposed by Smith et al. (2017), personalized FL arose from the observation that statistical heterogeneity of client data distributions can make training a single global model suboptimal (Kairouz et al., 2021). However, most works in personalized FL assume that all clients hold labeled data that they can use to obtain a personalized model, see Appendix C. A small number of works have looked at the problem of personalizing models to unlabeled clients. Ye et al. (2024) apply a popular test time adaptation method to a FL setting. A global predictive model is personalized on a client using gradient updates obtained with forward

*Table 3.* Accuracy of FLowDUP for architecture combinations on class-partitioned CIFAR10.

| Model Architecture | | Fraction Labeled ($p$) | | | |
|---|---|---|---|---|---|
| $h_1$ | $f$ | 0.1 | 0.2 | 0.5 | 1.0 |
| CNN | CNN | $66.1 \pm 1.5$ | $75.3 \pm 1.8$ | $83.3 \pm 1.5$ | $86.6 \pm 0.1$ |
| CNN | ResNet18 | $71.4 \pm 0.5$ | $80.0 \pm 3.3$ | $88.0 \pm 0.9$ | $90.7 \pm 0.2$ |
| ResNet18 | CNN | $67.0 \pm 2.4$ | $78.0 \pm 3.1$ | $87.3 \pm 1.3$ | $88.5 \pm 1.1$ |
| ResNet18 | ResNet18 | $72.7 \pm 1.3$ | $82.9 \pm 1.8$ | $91.0 \pm 0.4$ | $92.3 \pm 0.4$ |

*Table 4.* Accuracy of FLowDUP with different subspace dimensions, $k$, on class-partitioned CIFAR10.

| Fraction Labeled ($p$) | 0.1 | 0.2 | 0.5 | 1.0 |
|---|---|---|---|---|
| FLowDUP ($k = 200$) | $56.6 \pm 1.9$ | $65.7 \pm 2.9$ | $70.0 \pm 1.8$ | $71.2 \pm 0.7$ |
| FLowDUP ($k = 500$) | $60.0 \pm 1.9$ | $69.8 \pm 2.8$ | $75.9 \pm 1.2$ | $76.9 \pm 0.8$ |
| FLowDUP ($k = 2000$) | $64.5 \pm 0.3$ | $72.5 \pm 1.7$ | $80.3 \pm 1.5$ | $83.5 \pm 0.9$ |
| FLowDUP ($k = 10000$) | $66.1 \pm 1.5$ | $75.3 \pm 1.8$ | $83.3 \pm 1.5$ | $86.6 \pm 0.1$ |

*Table 5.* Comparison in the setting that clients have labeled data.

| Method | CIFAR10 | FEMNIST |
|---|---|---|
| Per-FedAvg | $69.0 \pm 2.5$ | $81.1 \pm 1.5$ |
| FedRep | $77.4 \pm 1.7$ | $82.8 \pm 0.7$ |
| pFedMe | $74.3 \pm 1.4$ | $86.1 \pm 0.4$ |
| kNN-Per | $80.8 \pm 1.5$ | $84.6 \pm 0.6$ |
| pFedHN | $60.9 \pm 2.4$ | $82.5 \pm 0.1$ |
| PeFLL | $88.9 \pm 0.6$ | $90.7 \pm 0.2$ |
| FLowDUP | $86.6 \pm 0.1$ | $89.3 \pm 0.1$ |

passes through an adaptation model. Closest to our own work are Amosy et al. (2024); Scott et al. (2024) which use a combination of an embedding network and hypernetwork to generate personalized models using only unlabeled data. Their approaches differ from ours in two key aspects. Firstly, while they are able to produce personalized models for unlabeled clients, they are not able to use the unlabeled clients during training. In contrast, FLowDUP also leverages unlabeled clients during training, which leads to improved performance. Secondly, and more importantly, they generate all the parameters of the personalized model, rather than a low-dimensional parameterization as FLowDUP does. Due to the high computational cost of this approach they are only able to generate very small personalized models for the clients. Moreover, even for such small models the hypernetwork is too large to send to the clients and instead remains on the server. This leads to complicated multi-step communication schemes that are inefficient and break the federated learning paradigm of working only with aggregate client statistics in order to maintain privacy. For example, in practice, methods such as secure aggregation are used, such that the server does not see individual client updates (Bonawitz et al., 2016; 2019; Ji et al., 2025). Approaches that require the server to access individual client updates are incompatible with such deployments. On the other hand, FLowDUP follows a standard FL protocol. Finally, several works have explored Federated Representation Learning in the presence of statistically heterogeneous clients (Zhang et al., 2020; Jang et al., 2022). These methods typically aim to learn a feature extractor(s) using unlabeled client data. However, obtaining a personalized predictive model on the client would still require the client to possess labels, and as such these methods are not applicable to the task we aim to solve.

**Theory** Beyond standard single-task learning, it has been shown that in *multi-task learning (MTL)* (Caruana, 1997; Baxter, 1995), when learning multiple related tasks, sharing information between them can provably improve the performance. Specifically, different works have studied the generalization behavior of multi-task learning (Maurer, 2006; Crammer & Mansour, 2012; Pontil & Maurer, 2013; Pentina & Lampert, 2017; Yousefi et al., 2018; Du et al., 2021) using notions like VC-dimensions (Vapnik & Chervonenkis, 1971) or Rademacher complexity (Bartlett & Mendelson, 2002). Recently, with the success of PAC-Bayesian bounds for studying generalization behavior of neural networks (Dziugaite & Roy, 2017), and due to their strength in parameterizing multi-task learning and meta-learning (Schmidhuber, 1987), PAC-Bayes multi-task learning has become an active line of research (Pentina & Lampert, 2014; Amit & Meir, 2018; Rothfuss et al., 2023; Guan & Lu, 2022; Zakerinia et al., 2024; Zakerinia & Lampert, 2025). However, the focus of these works is *inductive multi-task learning*, where all tasks have labeled data. In contrast in this work, we introduce *transductive multi-task learning*, where only a subset of the tasks have labeled data.

# 7. Conclusion

We presented FLowDUP, a method that can generate personalized models for clients using only a single forward pass with unlabeled data. FLowDUP's training objective is theoretically motivated by our new generalization bound in a transductive multi-task framework and is able to make use of both labeled and unlabeled clients during training for improved performance. Our empirical evaluation demonstrated that FLowDUP can achieve strong performance in the presence of both feature and label heterogeneity. In ablations, we investigated the effect of architecture choices, subspace dimension, the benefit of using unlabeled clients and how FLowDUP is able to capture dataset heterogeneity.

FLowDUP's main limitation is not a consequence of its algorithmic design, but of the problem it tries to solve: by definition, to create personalized models for clients with only unlabeled data, only properties of their *marginal* (input) data distribution can be exploited. Therefore, FLowDUP—like all other methods for this task—works well only when knowledge of the marginal (input data) distribution suffices to determine a good predictive model. In practice, this is indeed often the case, even for subjective prediction tasks, such as product or movie recommendations: e.g. the knowledge which movies a user has rented can suffice to determine their taste in movies, even without knowledge of explicit per-movie ratings.

# Impact Statement

This paper presents work whose goal is to advance the field of Machine Learning. There are many potential societal consequences of our work, none which we feel must be specifically highlighted here.

# Acknowledgements

This work was supported in part by the Austrian Science Fund (FWF) [10.55776/COE12]. This research was also supported by the Scientific Service Units (SSU) of ISTA through resources provided by Scientific Computing (Sci-Comp).

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

# A. Transductive multi-task learning

In this section, we provide our general theoretical contributions and their proof.

**PAC-Bayesian theory** (McAllester, 1998) studies the generalization behavior of stochastic models (a distribution over a set of models $f \in \mathcal{F}$). Formally, when training a posterior distribution $Q \in \mathcal{M}(\mathcal{F})$ using a dataset $S$ consisting of $m_L$ i.i.d. labeled samples from a distribution $D$ over $\mathcal{X} \times \mathcal{Y}$, PAC-Bayes bounds guarantee upper-bounds for the true risk of a posterior $Q$ i.e. $\mathcal{R}(Q) = \mathbb{E}_{f \sim Q} \mathbb{E}_{(x,y) \sim D} \ell(f, x, y)$ based on its training risk ($\widehat{\mathcal{R}}(Q) = \mathbb{E}_{f \sim Q} \frac{1}{m_L} \sum_{i=1}^{m_L} \ell(f, x_i, y_i)$), and a complexity term based on $\mathbf{KL}(Q \| P)$ where $P \in \mathcal{M}(\mathcal{F})$ is a data-independent prior over the space of models, and $\mathbf{KL}$ is the Kullback-Leibler divergence. As an example, from (Maurer, 2004) we have that for any prior $P$, and any $\delta > 0$, over the sampling of the dataset $S$ for all posteriors $Q$ it holds:

$$\mathcal{R}(Q) \le \widehat{\mathcal{R}}(Q) + \sqrt{\frac{\mathbf{KL}(Q \| P) + \log(\frac{2\sqrt{m_L}}{\delta})}{2m_L}}, \tag{8}$$

This bound holds in an inductive learning setting, i.e. we have a distribution $D$, and the goal is to generalize from training data to the unseen data. On the other hand, in transductive learning (Vapnik, 1998), there is a set of labeled data, and a set of unlabeled data, and the goal is to generalize from labeled to unlabeled data. Transductive learning has also been studied in the PAC-Bayes literature. Therefore, if we have $m$ samples, of which $m_L$ are labeled, we want to generalize from $\widehat{\mathcal{R}}(Q)$ to $\mathbb{E}_{f \sim Q} \frac{1}{m} \sum_{i=1}^{m} \ell(f, x_i, y_i)$. For this problem, (Bégin et al., 2014) have proved for any prior $P$, and any $\delta > 0$, over the sampling of $m_L$ labeled samples out of $m$ samples (without replacement) for all posteriors $Q$ it holds:

$$\mathbb{E}_{f \sim Q} \frac{1}{m} \sum_{i=1}^{m} \ell(f, x_i, y_i) \le \widehat{\mathcal{R}}_L(Q) + \sqrt{(1 - \frac{m_L}{m}) \frac{\mathbf{KL}(Q \| P) + \log(\frac{t(m_L, m)}{\delta})}{2m_L}}, \tag{9}$$

where $t(m_L, m)$ is the upper bound on the transductive combinatorial term of (Bégin et al., 2014). In the regime $20 \le m_L \le m - 20$, we have $t(m_L, m) = 3 \log(m_L) \sqrt{m_L(1 - \frac{m_L}{m})}$. Since the dominant term in both bounds is the $\mathbf{KL}$ terms, the transductive setting has an improved generalization guarantee by a factor of $\sqrt{1 - \frac{m_L}{m}}$.

**PAC-Bayes multi-task learning** Beyond standard single-task learning, it has been shown theoretically that when learning multiple related tasks, sharing information between them can improve performance. Formally, in *multi-task learning (MTL)* (Caruana, 1997), there are $n$ tasks with different distributions $D_1, \ldots, D_n$ and respective datasets $S_1, \ldots, S_n$, and the goal is to learn individual models (or Posteriors) jointly. *Meta-learning* (Schmidhuber, 1987), or *learning to learn* (Thrun & Pratt, 1998), extends multi-task learning to the setting where there will also be future tasks that are not observed yet, with the assumption that the observed tasks and future tasks are all i.i.d. samples from a distribution $\tau$ over an environment of tasks (Baxter, 2000). In multi-task learning, the goal is to generalize from the average training error to the average true risk of observed clients, and in meta-learning, the goal is to generalize to the expected true risk over $\tau$. Given that in the training step, we cannot learn a model for future tasks, meta-learning is formalized as learning an algorithm to apply to a future task. However, past works focused on multi-task learning and meta-learning in an inductive way, i.e. the model suggested by Baxter (2000). In this work, we focus on a transductive version where there are $n$ tasks, of which $n_L$ tasks are selected randomly to have a labeled dataset, and the remaining $n - n_L$ tasks have only unlabeled data.

Following Pentina & Lampert (2014), PAC-Bayes is a popular framework to study multi-task learning and meta-learning, given its ability to share information between tasks through the concept of a prior. In this work, we follow the framework introduced in Zakerinia et al. (2024). As our theoretical contribution, we prove new bounds for the transductive multi-task learning scenario we described.

Formally, we have access to $n_L$ labeled datasets and $n - n_L$ unlabeled datasets, and our goal is to generate models with good performance on all tasks. Since we have unlabeled tasks, we work with a family of algorithms $\mathcal{A}$ that can create posteriors from unlabeled data i.e. $A : \mathcal{P}(\mathcal{X}) \to \mathcal{M}(\mathcal{F})$ is a mapping from the set of unlabeled datasets to models. We aim to learn a stochastic algorithm i.e. a *meta-posterior* over a set of algorithms ($\rho \in \mathcal{M}(\mathcal{A})$) that have good performance on all tasks. For each algorithm, $A(X_1), \ldots, A(X_n)$ are the task posteriors, and we denote to $\mathcal{Q}(A)$ as a hyper-posterior, a distribution over priors to capture the similarities between the outputs of the algorithm, and to have a data-dependent prior for our PAC-Bayes bounds. For a detailed explanation of the role of hyper-posterior, we refer the reader to Zakerinia et al. (2024).

For each meta-posterior, our goal is to minimize the average true risk of all tasks:

$$\mathcal{R}(\rho) = \mathbb{E}_{A\sim\rho} \frac{1}{n} \sum_{i=1}^{n} \mathbb{E}_{(x_i,y_i)\sim\mathcal{D}_i} \mathbb{E}_{f\sim A(X_i)} \ell(y_i, f(x_i)) \tag{10}$$

However, the true distribution of tasks is unknown, and we cannot compute the training risk of unlabeled clients; therefore, the computable object is the average training risk of labeled clients:

$$\widehat{\mathcal{R}}(\rho) = \mathbb{E}_{A\sim\rho} \frac{1}{n_L} \sum_{i=1}^{n_L} \frac{1}{m} \sum_{j=1}^{m} \mathbb{E}_{f\sim A(X_i)} \ell(y_{i,j}, f(x_{i,j})) \tag{11}$$

We now state our main theoretical results:

**Theorem A.1.** *For any fixed meta-prior $\pi$, fixed hyper-prior $\mathcal{P}$ and any $\delta > 0$ with probability at least $1-\delta$ over the sampling of the datasets, for all distributions $\rho \in \mathcal{M}(\mathcal{A})$ over algorithms, and for all hyper-posterior functions $\mathcal{Q} : \mathcal{A} \to \mathcal{M}(\mathcal{M}(\mathcal{F}))$ it holds*

$$\mathcal{R}(\rho) \leq \widehat{\mathcal{R}}(\rho) + \sqrt{(1 - \frac{n_L}{n}) \frac{\mathbf{KL}(\rho\|\pi) + \log(\frac{2t(n_L,n)}{\delta})}{2n_L}} + \mathbb{E}_{A\sim\rho} \sqrt{\frac{C(A,\mathcal{Q},\mathcal{P}) + \log(\frac{8mn}{\delta}) + 1}{2mn}} \tag{12}$$

*where,*

$$C(A,\mathcal{Q},\mathcal{P}) = \mathbf{KL}(\mathcal{Q}(A)\|\mathcal{P}) \quad + \mathbb{E}_{P\sim\mathcal{Q}(A)} \sum_{i=1}^{n} \mathbf{KL}(A(S_i)\|P) \tag{13}$$

*Proof.* For the proof, we define the following intermediate objective, which quantifies the training risk of all clients. Note that this object exists (There is a labeling for the unlabeled data, however, we do not have access to them, i.e. the loss function for unlabeled clients is well-defined, but we cannot compute it.)

$$\widetilde{\mathcal{R}}(\rho) = \mathbb{E}_{A\sim\rho} \frac{1}{n} \sum_{i=1}^{n} \frac{1}{m} \sum_{j=1}^{m} \mathbb{E}_{f\sim A(X_i)} \ell(y_{i,j}, f(x_{i,j})) \tag{14}$$

We divide the proof into bounding $\mathcal{R}(\rho) - \widetilde{\mathcal{R}}(\rho)$, and bounding $\widetilde{\mathcal{R}}(\rho) - \widehat{\mathcal{R}}(\rho)$ separately. For stating our results, we use the following notations as in Zakerinia et al. (2024):

- *Posterior $\mathfrak{Q}(A, \mathcal{Q}(A))$:* given as input an algorithm $A \in \mathcal{A}$ and a hyper-posterior mapping $\mathcal{Q} : \mathcal{A} \to \mathcal{M}(\mathcal{M}(\mathcal{F}))$ as input, it outputs the distribution over $\mathcal{M}(\mathcal{F}) \times \mathcal{F}^{\otimes n}$ with the following generating process: *i)* sample a prior $P \sim \mathcal{Q}(A)$, *ii)* for each task, $i = 1, \ldots, n$, sample a model $f_i \sim A(X_i)$.

- *Prior $\mathfrak{P}$:* For the hyper-prior $\mathcal{P} \in \mathcal{M}(\mathcal{F})$ as input, it outputs the distribution over $\mathcal{M}(\mathcal{F}) \times \mathcal{F}^{\otimes n}$ with the following generating process: *i)* sample a prior $P \sim \mathcal{P}$, *ii)* for each task, $i = 1, \ldots, n$, sample a model $f_i \sim P$.

The first part is bounding $\mathcal{R}(\rho) - \widetilde{\mathcal{R}}(\rho)$ i.e. multi-task generalization bound for all $n$ tasks. The change of objective and intermediate step we have here compared to Zakerinia et al. (2024) allows us to consider the generalization behavior of all tasks (labeled and unlabeled) in contrast to only the labeled tasks. For the proof, for any task $i$ and any model $f_i$ we define:

$$\Delta_i(f_i) = \mathbb{E}_{x\sim D_i} \ell(y, f_i(x)) - \frac{1}{m} \sum_{j=1}^{m} \ell(y_{i,j}, f_i(x_{i,j})) \tag{15}$$

By this definition and the definitions of $\widetilde{\mathcal{R}}$ and $\mathcal{R}$ we have:

$$\mathbb{E}_{(P,f_1,f_2,\ldots,f_n)\sim\mathfrak{Q}(A,\mathcal{Q}(A))} \left[ \frac{1}{n} \sum_{i=1}^{n} \Delta_i(f_i) \right] = \mathcal{R}(A) - \widetilde{\mathcal{R}}(A) \tag{16}$$

By *change of measure inequality* (Seldin et al., 2012), for any $\lambda > 0$, any $A \in \mathcal{A}$ and any $\mathcal{Q} : \mathcal{A} \to \mathcal{M}(\mathcal{M}(\mathcal{F}))$, we have:

$$\mathcal{R}(A) - \widetilde{\mathcal{R}}(A) - \frac{1}{\lambda} \mathbf{KL}\left(\mathfrak{Q}(A, \mathcal{Q}(A)) \| \mathfrak{P}\right) \leq \frac{1}{\lambda} \log \mathbb{E}_{(P, f_1, f_2, \ldots, f_n) \sim \mathfrak{P}} \prod_{i=1}^{n} e^{\frac{\lambda}{n} \Delta_i(f_i)} \tag{17}$$

Because $\mathfrak{P}$ is independent of $S_1, \ldots, S_n$, we have

$$\mathbb{E}_{S_1, \ldots, S_n} \mathbb{E}_{(P, f_1, f_2, \ldots, f_n) \sim \mathfrak{P}} \prod_{i=1}^{n} e^{\frac{\lambda}{n} \Delta_i(f_i)} = \mathbb{E}_{P \sim \mathcal{P}} \prod_{i=1}^{n} \mathbb{E}_{S_i} \mathbb{E}_{f_i \sim P} e^{\frac{\lambda}{n} \Delta_i(f_i)} \tag{18}$$

By Hoeffding's lemma for $\Delta_i(f_i)$, we have

$$\mathbb{E}_{S_i} \mathbb{E}_{f_i \sim P} e^{\frac{\lambda}{n} \Delta_i(f_i)} \leq e^{\frac{\lambda^2}{8n^2 m}}. \tag{19}$$

Therefore by combining (18) and (19) we have:

$$\mathbb{E}_{S_1, \ldots, S_n} \mathbb{E}_{(P, f_1, f_2, \ldots, f_n) \sim \mathfrak{P}} \prod_{i=1}^{n} e^{\frac{\lambda}{n} \Delta_i(f_i)} \leq e^{\frac{\lambda^2}{8nm}}. \tag{20}$$

By Markov's inequality, for any $\epsilon > 0$ we have

$$\mathbb{P}_{S_1, \ldots, S_n} \left( \mathbb{E}_{(P, f_1, f_2, \ldots, f_n) \sim \mathfrak{P}} \prod_{i=1}^{n} e^{\frac{\lambda}{n} \Delta_i(f_i)} \geq e^{\epsilon} \right) \leq e^{\frac{\lambda^2}{8nm} - \epsilon} \tag{21}$$

Hence by combining (17) and (21) we get

$$\mathbb{P}_{S_1, \ldots, S_n} \left( \exists A, \mathcal{Q} : \mathcal{R}(A) - \widetilde{\mathcal{R}}(A) - \frac{1}{\lambda} \mathbf{KL}(\mathfrak{Q}(A, \mathcal{Q}(A)) \| \mathfrak{P}) \geq \frac{1}{\lambda} \epsilon \right) \leq e^{\frac{\lambda^2}{8nm} - \epsilon}, \tag{22}$$

or, equivalently, it holds for any $\delta > 0$ with probability at least $1 - \frac{\delta}{2}$:

$$\forall A, \mathcal{Q} : \quad \mathcal{R}(A) - \widetilde{\mathcal{R}}(A) \leq \frac{1}{\lambda} \mathbf{KL}(\mathfrak{Q}(A, \mathcal{Q}(A)) \| \mathfrak{P}) + \frac{1}{\lambda} \log(\frac{2}{\delta}) + \frac{\lambda}{8nm} \tag{23}$$

With a union bound for $\lambda \in \{1, \ldots, 4mn\}$, and choosing the best $\lambda$ we get:

$$\mathbb{P}_{S_1, \ldots, S_n} \left( \forall A, \mathcal{Q} : \mathcal{R}(A) - \widetilde{\mathcal{R}}(A) \leq \sqrt{\frac{\mathbf{KL}\left(\mathfrak{Q}(A, \mathcal{Q}(A)) \| \mathfrak{P}\right) + \log(\frac{8mn}{\delta}) + 1}{2mn}} \right) \geq 1 - \frac{\delta}{2} \tag{24}$$

With probability at least $1 - \frac{\delta}{2}$, the bound holds for all $A \in \mathcal{A}$, therefore, we can take the expectation for $A \in \rho$. Given that $\mathbb{E}_{A \sim \rho}[\mathcal{R}(A) - \widetilde{\mathcal{R}}(A)] = \mathcal{R}(\rho) - \widetilde{\mathcal{R}}(\rho)]$, the following holds with probability at least $1 - \frac{\delta}{2}$, for all $\rho$, and $\mathfrak{Q}$:

$$\forall \rho, \mathcal{Q} : \mathcal{R}(\rho) - \widetilde{\mathcal{R}}(\rho) \leq \mathbb{E}_{A \sim \rho} \sqrt{\frac{\mathbf{KL}(\mathfrak{Q}(A, \mathcal{Q}(A)) \| \mathfrak{P}) + \log(\frac{8mn}{\delta}) + 1}{2mn}}. \tag{25}$$

For $\mathbf{KL}(\mathfrak{Q}(A, \mathcal{Q}(A)) \| \mathfrak{P})$, we have:

$$\begin{aligned}
\mathbf{KL}(\mathfrak{Q}(A, \mathcal{Q}(A)) \| \mathfrak{P}) &= \mathbb{E}_{P \sim \mathcal{Q}(A)} \left[ \mathbb{E}_{f_i \sim A(X_i)} \ln \frac{\mathcal{Q}(A)(P) \prod_{i=1}^{n} A(X_i)(f_i)}{\mathcal{P}(P) \prod_{i=1}^{n} P(f_i)} \right] \\
&= \mathbb{E}_{P \sim \mathcal{Q}(A)} \left[ \ln \frac{\mathcal{Q}(A)(P)}{\mathcal{P}(P)} \right] + \mathbb{E}_{P \sim \mathcal{Q}(A)} \left[ \sum_{i=1}^{n} \mathbb{E}_{f_i \sim A(S_i)} \ln \frac{A(X_i)(f_i)}{P(f_i)} \right] \\
&= \mathbf{KL}(\mathcal{Q}(A) \| \mathcal{P}) + \mathbb{E}_{P \sim \mathcal{Q}(A)} \sum_{i=1}^{n} \mathbf{KL}(A(S_i) \| P) \tag{26}
\end{aligned}$$

Combining equations (25) and (26) gives that with probability $1 - \frac{\delta}{2}$, for all $\rho \in \mathcal{M}(\mathcal{A}), \mathcal{Q} : \mathcal{A} \to \mathcal{M}(\mathcal{M}(\mathcal{F}))$:

$$\mathcal{R}(\rho) - \widetilde{\mathcal{R}}(\rho) \leq \underset{A \sim \rho}{\mathbb{E}} \sqrt{\frac{C(A, \mathcal{Q}, \mathcal{P}) + \log(\frac{8mn}{\delta}) + 1}{2mn}} \tag{27}$$

For the second part, we apply the transductive bound (9) to generalization between $\widetilde{\mathcal{R}}(\rho)$ and $\widehat{\mathcal{R}}(\rho)$, we get:

$$\widetilde{\mathcal{R}}(\rho) \leq \widehat{\mathcal{R}}(\rho) + \sqrt{(1 - \frac{n_L}{n}) \frac{\mathbf{KL}(\rho || \pi) + \log(\frac{2t(n_L, n)}{\delta})}{2n_L}} \tag{28}$$

Combining (27) and (28) proves the theorem. $\qquad \square$

From this general theorem, we can now prove the generalization bound for our algorithm FLowDUP.

**Theorem 4.1.** *For all $\delta > 0$, and any loss function $\ell : \mathcal{Y} \times \mathcal{Y} \to [0, 1]$, the following statement holds with probability at least $1 - \delta$ over the sampling of $n_L$ clients of $n$ clients and randomness of the dataset. For all parameter vectors, $\psi = (\psi_h, \psi_r)$:*

$$\mathcal{R}(\rho_h) \leq \widehat{\mathcal{R}}(\rho_h) + \sqrt{\left(1 - \frac{n_L}{n}\right) \frac{\frac{1}{2\alpha_h} \|\psi_h\|^2 + c_1}{2n_L}} + \tag{7}$$

$$\underset{\psi_h' \sim \rho_h}{\mathbb{E}} \sqrt{\frac{\frac{1}{2\alpha_\theta} \sum_{i=1}^n \mathbb{E}_{\psi_r'} c_i(h, \psi_h', \psi_r') + \frac{1}{2\alpha_r} \|\psi_r\|^2 + c_2}{2mn}}$$

*where $c_i(h, \psi_h', \psi_r') = \|h(S_i \, ; \, \psi_h') - \psi_r'\|^2$, and $c_1$ and $c_2$ are logarithmic terms in $n$ and $n_L$.*

*Proof.* For each hypernetwork with trainable parameters $\psi_h$, consider an algorithm $A$ that for task $i$ generates $A(S_i) = \mathcal{N}(h(S_i \, ; \, \psi_h) \, ; \, \alpha_\theta \mathrm{Id})$.

For each $v \in \mathbb{R}^k$, the corresponding prior is $P = \mathcal{N}(v; \alpha_\theta \mathrm{Id})$, and $\mathcal{P} = \mathcal{N}(0; \alpha_r \mathrm{Id})$ is the hyper-prior over priors i.e. over their mean. Define the meta-prior as $\pi = \mathcal{N}(0; \alpha_h \mathrm{Id})$ over $\mathcal{A}$ i.e. over hypernetwork parameters $\psi_h$.

Define the meta-posterior as a Gaussian distribution over the parameters of the hypernetwork, $\rho_h = \mathcal{N}(\psi_h \, ; \, \alpha_h \mathrm{Id})$ for a fixed $\alpha_h$. Define the hyper-posterior as $\mathcal{Q} = \mathcal{N}(\psi_r \, ; \, \alpha_r \mathrm{Id})$.

By applying the theorem A.1 to the defined distributions, we complete the proof. $\qquad \square$

# B. Experiments

Here we include additional experiments (Section B.1) and implementation details (Section B.2).

## B.1. Additional Experiments

**Rotated MNIST**     We include here results for Rotated MNIST. The observations here are broadly similar to those discussed in Section 5.2 with FLowDUP exhibiting the best performance overall.

*Table 6.* Accuracy on Rotated MNIST for varying fractions ($p$) of the training clients having labeled data.

| Fraction Labeled ($p$) | 0.1 | 0.2 | 0.5 | 1.0 |
|---|---|---|---|---|
| FedAvg | $92.9 \pm 0.2$ | $94.8 \pm 0.2$ | $96.0 \pm 0.2$ | $96.4 \pm 0.1$ |
| FedProx | $92.8 \pm 0.4$ | $94.7 \pm 0.1$ | $95.8 \pm 0.2$ | $96.3 \pm 0.1$ |
| LD-FedAvg | $90.5 \pm 0.3$ | $92.4 \pm 0.3$ | $94.1 \pm 0.0$ | $95.0 \pm 0.1$ |
| FedTTA | $93.2 \pm 0.3$ | $94.8 \pm 0.1$ | $96.6 \pm 0.1$ | $97.3 \pm 0.1$ |
| FLowDUP | $94.0 \pm 1.4$ | $96.6 \pm 0.4$ | $97.8 \pm 0.1$ | $98.5 \pm 0.1$ |

**The effect of unlabeled clients**     One of the advantages of FLowDUP is that its learning objective is structured in such a way that it allows unlabeled clients that are present during training to contribute to regularizing $h$. Specifically, unlabeled

clients are able to compute the regularizer $\Omega$ in Equation (6) and obtain from this a gradient update for $\psi$. Here we investigate the effect that unlabeled clients have. To do this we run FLowDUP on partitioned CIFAR10 both with and without using the unlabeled clients. For FLowDUP without unlabeled clients we set the cohort size to 100 and sample only clients with labeled data. For FLowDUP with unlabeled clients we set the cohort size to 200 and the labeled client sampling rate to $\alpha = 0.5$ so that the number of labeled clients present during each round is the same in both cases. We vary the total fraction of labeled clients, with $p \in \{0.05, 0.1, 0.2, 0.5\}$. As always we set $k = 10^4$. Tables 7 and 8 show the results. As we can see, overall using unlabeled clients leads to an increase in performance. This is most pronounced for lower values of $p$ (in particular at $p = 0.5$ the effect is small or negligible) and occurs for both architectures.

*Table 7.* Partitioned CIFAR10 Accuracy for CNN.

| Fraction Labeled ($p$) | 0.05 | 0.1 | 0.2 | 0.5 |
|---|---|---|---|---|
| FLowDUP (without unlabeled) | $51.5 \pm 1.8$ | $65.3 \pm 1.3$ | $74.5 \pm 1.3$ | $82.5 \pm 1.0$ |
| FLowDUP (with unlabeled) | $52.3 \pm 1.4$ | $67.5 \pm 0.4$ | $76.6 \pm 2.3$ | $83.8 \pm 0.7$ |

*Table 8.* Partitioned CIFAR10 Accuracy for ResNet18.

| Fraction Labeled ($p$) | 0.05 | 0.1 | 0.2 | 0.5 |
|---|---|---|---|---|
| FLowDUP (without unlabeled) | $54.8 \pm 1.3$ | $71.7 \pm 1.2$ | $83.3 \pm 3.5$ | $90.5 \pm 0.4$ |
| FLowDUP (with unlabeled) | $57.5 \pm 1.6$ | $72.6 \pm 0.6$ | $83.0 \pm 2.7$ | $90.6 \pm 0.7$ |

**The learnable regularizer** FLowDUP prevents overfitting by penalizing large deviations between the generated client subspace parameters and a learned regularizer $\psi_r$. Intuitively, we can think of $\psi_r$ as a global subspace model that clients should not deviate too much from. Through the lens of our theoretical results, (Theorem 4.1), $\psi_r$ is the mean of a learnable prior distribution on the client models. Here we investigate the efficacy of *learning* the regularizer. To do this we replace $\psi_r$ by 0, so that the regularization term in the loss becomes instead

$$\Omega = \sum_{i \in \mathcal{C}} \|h(X_i \,;\, \psi_h)\|^2. \tag{29}$$

Note that this is of course still a reasonable regularizer more in line with classic $\ell_2$ regularization. We train FLowDUP using this new non-learnable regularizer and compare it to our proposed version using the learnable regularizer in Equation (6). As in the previous section we again take a cohort size of 100 labeled clients and 100 unlabeled clients each round, i.e. $\alpha = 0.5$. We set $k = 10^4$ and vary $p \in \{0.1, 0.2, 0.5, 1.0\}$. The results can be seen in Table 9. They indeed show that the extra flexibility afforded by learning the regularizer leads to a modest boost in performance for all values of $p$.

**Labeled Client Sampling Rate** The hyperparameter $\alpha$ is introduced in order to fix the fraction of clients with labels participating per round. This is important for controlling the amount of labeled signal per round, which is particularly relevant when the number of unlabeled clients is much larger than the number of labeled clients. If we sampled only (or nearly only) unlabeled clients our hypernetwork would not learn to generate good models. Here we include an ablation study testing the influence of $\alpha$. We do this in the setting of CIFAR10 with CNN and $p = 0.2$ from Table 1. The results are shown in Table 10. As we can see, performance decreases if $\alpha$ is chosen too low and plateaus quite quickly as we increase it.

### B.2. Implementation Details

**Hyperparameters** There are a number of general federated learning hyperparameters that are shared across methods. We train all methods for the same number of global rounds $T$. For class partitioned CIFAR10 and FEMNIST we set $T = 1000$ while for Rotated Fashion-MNIST and Rotated MNIST we set $T = 500$. All methods use a client cohort size each round of size 100, a local number of epochs set to $E = 1$ and a local batch size of $B = 20$ for FEMNIST and $B = 50$ on all other datasets. For all methods we tune the local learning rate $\eta_l$ on validation clients.

Regarding method specific hyperparameters. For FedProx, we set $\mu$ following Li et al. (2020), that is $\mu = 1$ initially and is updated over the course of training as described in Li et al. (2020). For FedTTA we tune the adaptive learning rate. For FLowDUP, unless otherwise stated, we set the labeled client sampling rate to $\alpha = 0.9$ and the subspace dimension to $k = 10^4$. We tune the regularization strength $\lambda$.

*Table 9.* The effect of learning the regularizer $\psi_r$. Accuracy on partitioned CIFAR10 for CNN.

| Fraction Labeled ($p$) | 0.1 | 0.2 | 0.5 | 1.0 |
|---|---|---|---|---|
| FLowDUP (Without Learnable Reg) | $65.5 \pm 0.4$ | $75.2 \pm 0.8$ | $83.3 \pm 1.7$ | $85.6 \pm 0.8$ |
| FLowDUP (With Learnable Reg) | $67.5 \pm 0.4$ | $76.6 \pm 0.3$ | $83.8 \pm 0.7$ | $86.6 \pm 0.1$ |

*Table 10.* The effect of the labeled client sampling rate $\alpha$. Accuracy on partitioned CIFAR10 for CNN.

| $\alpha$ | 0.1 | 0.2 | 0.5 | 0.8 | 0.9 |
|---|---|---|---|---|---|
| Accuracy | $71.4 \pm 2.8$ | $74.4 \pm 1.1$ | $75.0 \pm 3.2$ | $75.6 \pm 1.5$ | $75.3 \pm 1.8$ |

**Model architectures**    When used for prediction the CNN follows the architecture used in prior work (McMahan et al., 2017) while the ResNet18 follows the standard architecture with the final classification layer having output dimension 10 for the 10 classes present in CIFAR10. When used for $h_1$, we instead replace the final linear layers of the CNN and ResNet18 with another with output dimension 256. For $h_2$ we always use a fully connected network with a single hidden layer and ReLU non-linearity.

**Efficient random expansion.**    To efficiently implement random projection $P$, we use the Kronecker product projector of Lotfi et al. (2022), $P = Q_1 \otimes Q_2 / \sqrt{D}$, for $Q_1, Q_2 \sim \mathcal{N}(0,1)^{\sqrt{D} \times \sqrt{d}}$. By this construction, the matrix $P \in \mathbb{R}^{D \times d}$ never has to be explicitly instantiated, which requires memory and computation in $O(\sqrt{dD})$ instead of $O(dD)$.

# C. Additional Related Work

**Personalized federated learning**    Approaches to personalized FL typically fall into one of the following categories: Meta-learning based approaches (Jiang et al., 2019; Fallah et al., 2020) which learn a single global model that can be easily personalized using a small number of gradient steps on the client. Parameter decomposition-approaches (Arivazhagan et al., 2019; Collins et al., 2021; Marfoq et al., 2022; Chen et al., 2023; Wu et al., 2023) which divide the learnable parameters into some that are shared across clients (such as a feature extractor) and some that are specific to each individual client (such as a classification head). Federated multi-task approaches (Smith et al., 2017; Dinh et al., 2020; Hanzely et al., 2020; Marfoq et al., 2021; Li et al., 2021; Lin et al., 2022; Ye et al., 2023; Zhang et al., 2023) learn separate models for each client while still sharing some information across clients, for instance by regularizing towards some global model. All of the above approaches, however, require a client to have labeled data in order to obtain a personalized model.

**Learning in a subspace**    This formulation of intrinsic dimensionality and learning in a subspace has been studied in different contexts in the literature. (Li et al., 2018) introduced the formulation and showed that for real-world problems the intrinsic dimension is much smaller than the total number of parameters $k \ll d$. (Aghajanyan et al., 2021) showed that pretraining reduces the intrinsic dimensionality of fine-tuning. (Lotfi et al., 2022) used this parametrization to achieve non-vacuous generalization bounds for neural networks. (Zakerinia et al., 2025) extended the definition of the intrinsic dimension to multi-task learning. (Park et al., 2025) used the formulation for black-box prompt tuning. (Lin et al., 2022) used it to reduce the communication of the global model in personalized federated learning.

