# OpenReview forum: "Federated Learning with Unlabeled Clients: Personalization Can Happen in Low Dimensions"
_ICML.cc/2026/Conference — ICML 2026 regular_

### Official Review · Reviewer_qtcz · 2026-02-25

**Soundness:** 4
**Presentation:** 4
**Significance:** 4
**Originality:** 4
**Overall Recommendation:** 5
**Confidence:** 5

**Summary:**

This paper proposes FLowDUP, a personalized FL method that generates client-specific models using only unlabeled data via a small hypernetwork that outputs a low-dimensional subspace parameterization of the model, later expanded to full dimension through a fixed random projection. A new transductive multi-task PAC-Bayesian generalization bound motivates the training objective: a labeled loss computed on labeled clients and an unlabeled regularizer that uses all clients, enabling unlabeled clients to contribute during training. Extensive experiments on CIFAR-10, Rotated MNIST/Fashion-MNIST, and FEMNIST show consistent gains over applicable baselines, along with ablations on subspace size and architecture choices.

**Compliance With Llm Reviewing Policy:**

Affirmed.

**Final Justification:**

My concerns have been addressed.

**Key Questions For Authors:**

1. How many unlabeled samples are used at inference to form $r(X)$? How sensitive is performance to that number, especially for clients with very small datasets?

**Limitations:**

Yes

**Strengths And Weaknesses:**

## Strengths

1. The paper addresses an important challenge, i.e., personalization for unlabeled clients in FL, which is relevant in privacy-preserving, low-participation scenarios where label acquisition is costly.
2. It introduces a practical hypernetwork-in-the-loop approach for unlabeled personalization that outputs only low-dimensional parameters, making on-device generation feasible while preserving standard FL constraints.
3. It also derives a transductive multi-task PAC-Bayesian bound tailored to the setting with mixed labeled and unlabeled clients. The bound justifies the objective structure and the learnable prior regularizer that unlabeled clients can influence.
4. Evaluations across diverse sources of heterogeneity over both label- and feature-level show consistent performance, with multiple architectures and varying labeled fractions.

## Weaknesses

1. The paper’s central practical claim, i.e, on-device feasibility via low-dimensional parameterization, is not backed by concrete systems metrics, such as communication payload per round or server aggregation cost.
2. No study on how many unlabeled samples are needed at inference to generate an effective personalized model.

---

> ### Author Rebuttal · Authors · 2026-03-30
>
> Thank you for your review and the strong positive assessment of our work. We hope you will continue to support our contribution during the upcoming discussion and decision phase.
>
> > The paper’s central practical claim, i.e, on-device feasibility via low-dimensional parameterization, is not backed by concrete systems metrics, such as communication payload per round or server aggregation cost.
>
> Note that -as it is common for FL work at ICML- our study is simulation-based, so we cannot report real deployment latency/bandwidth. Using parameter dimension / memory size as a proxy for communication, we report key efficiency gains already in the appendix: e.g. with $k=10^4$, for a ResNet18-scale model, the hypernetwork output layer shrinks from about 3.07B weights (12.3 GB in fp32) to about 2.56M weights (about 10 MB). Note that the random expansion and initialization can be generated locally from a shared seed and thus need not be transmitted. We will move these system-facing numbers into the main text in the revision.
>
>
> > No study on how many unlabeled samples are needed at inference to generate an effective personalized model. [...] How many unlabeled samples are used at inference to form $r(X)$? How sensitive is performance to that number, especially for clients with very small datasets?
>
> Thank you for the suggestion. We did not explore this direction, because we found that already one batch of samples (20 for all datasets, except 50 for FEMNIST) sufficed to reliably produce good models. But indeed, it would be interesting to explore (theoretically and empirically) if this value can be reduced even more.

---

> > ### Author Rebuttal · Reviewer_qtcz · 2026-03-31
> >
> > Thank you for the clarifications, and I believe my current score is appropriate.

---

### Official Review · Reviewer_Xaig · 2026-03-11

**Soundness:** 3
**Presentation:** 3
**Significance:** 3
**Originality:** 3
**Overall Recommendation:** 4
**Confidence:** 3

**Summary:**

This paper studies the problem of personalized federated learning when some clients do not have labeled data. Most existing personalized FL methods assume that each client has access to labeled data for local training or fine-tuning, which limits their applicability in realistic settings where some clients may only possess unlabeled data.

To address this limitation, the paper proposes FLowDUP, a method that generates personalized client models using only a forward pass on unlabeled data. The key idea is to restrict personalized model parameters to lie in a low-dimensional subspace, which enables efficient communication and computation. The method is theoretically motivated by a transductive multi-task PAC-Bayesian generalization bound, which provides guarantees for clients with unlabeled data. The training objective is designed so that both labeled and unlabeled clients can participate in the learning process.

The paper also provides empirical evaluation across several datasets with heterogeneous client distributions, along with ablation studies analyzing different components of the proposed method.

**Compliance With Llm Reviewing Policy:**

Affirmed.

**Final Justification:**

The rebuttal addresses some concerns, but the limited novelty relative to existing personalized FL literature remains. I maintain my original score of 4 (Weak Accept).

**Key Questions For Authors:**

1. Could the authors provide more intuition on how unlabeled clients contribute to learning the personalization subspace? In particular, how does the method avoid degenerate solutions when large fractions of clients have no labels?

2. How does the method perform when the proportion of unlabeled clients increases significantly (e.g., 70–90%)? Additional experiments varying this ratio would help understand the robustness of the approach.

3. How does the proposed method scale with a large number of clients and higher-dimensional models? Are there communication or optimization challenges in such scenarios?

4. Could the authors compare their approach with more recent parameter-efficient personalization methods in federated learning?

5. Is there any interpretation or analysis of the learned low-dimensional parameter space? For example, does it capture meaningful client variation?

**Limitations:**

yes

**Strengths And Weaknesses:**

### Strengths

1. **Addresses an important and realistic FL scenario**

The paper studies personalized federated learning with unlabeled clients, which is a practical and underexplored scenario. In real-world deployments, many edge clients may not have labeled data available, and enabling them to benefit from personalization is an important problem.

2. **Interesting low-dimensional personalization formulation**

Constraining personalized model parameters to lie in a low-dimensional subspace is an elegant idea. This design can reduce communication overhead and help mitigate overfitting when labeled data is scarce.

3. **Theoretical grounding**

The work introduces a transductive multi-task PAC-Bayesian generalization bound that motivates the proposed training objective. Providing theoretical insight into how unlabeled clients can participate in federated learning strengthens the methodological foundation of the work.

4. **Empirical evaluation with ablations**

The paper includes experiments across multiple datasets and provides ablation studies analyzing the role of different components of the method.

### Weaknesses

1. **Limited novelty relative to existing personalized FL literature**

While the paper addresses unlabeled clients, the overall framework is related to several existing lines of work, including low-rank or subspace-based personalization and parameter-efficient personalization methods. The novelty mainly lies in the specific formulation and theoretical motivation.

2. **Limited clarity on the role of unlabeled data**

The paper could more clearly explain how unlabeled data contributes to learning the personalization subspace and how personalization quality emerges without labeled supervision. Additional discussion or experiments varying the proportion of unlabeled clients would strengthen the contribution.

3. **Experimental scope could be broader**

The empirical evaluation is reasonable but somewhat limited. It would be helpful to see experiments in larger-scale federated settings with more clients, as well as comparisons with additional recent personalization approaches.

4. **Theoretical section may be difficult to follow**

Although the theoretical analysis is interesting, parts of the presentation are quite dense and may be difficult for readers to follow. Additional intuition or explanations would improve clarity.

---

> ### Author Rebuttal · Authors · 2026-03-30
>
> Thank you for your review and the positive assessment of our work. We hope you will continue to support our contribution during the upcoming discussion and decision phase.
>
>
> >  Limited novelty relative to existing personalized FL literature: [...] The novelty mainly lies in the specific formulation and theoretical motivation.
>
> We agree that the novelty of our work lies in the specific formulation of the FlowDUP algorithm, its foundations in theory, as (additionally) the strong empirical performance. We do not consider these as weaknesses, though.
>
> > Limited clarity on the role of unlabeled data: The paper could more clearly explain how unlabeled data contributes to learning the personalization subspace and how personalization quality emerges without labeled supervision. Additional discussion or experiments varying the proportion of unlabeled clients would strengthen the contribution.
>
> This seems to be a misunderstanding: the personalization subspace is provided by a fixed random matrix (see e.g. lines 039 and 077, right column). The effect of different proportions of unlabeled clients we illustration in our ablation studies (Tables 7, 8 and 9).
>
> > Experimental scope could be broader: The empirical evaluation is reasonable but somewhat limited. It would be helpful to see experiments in larger-scale federated settings with more clients, as well as comparisons with additional recent personalization approaches.
>
> Regarding the experimental settings, please see our response to Reviewer qjJJ. We follow standard practice from the literature. Regarding other baselines, please see our response to Reviewer qh3s. We report results for all existing pFL methods we are aware of that can create models for future clients without labels, plus FedAvg as a sanity check.
>
> However, if there are specific benchmark datasets or baseline methods you have in mind, please let us know so we can try them.
>
> > Theoretical section may be difficult to follow: Although the theoretical analysis is interesting, parts of the presentation are quite dense and may be difficult for readers to follow. Additional intuition or explanations would improve clarity.
>
> Indeed, because of the page limit, we had to keep the theory section in the main body rather short and dense. However, we do provide a high-level overview in the “Discussion” after Theorem 4.1. For a more accessible step-by-step derivation, please see our Appendix A.
>
> > Could the authors provide more intuition on how unlabeled clients contribute to learning the personalization subspace? In particular, how does the method avoid degenerate solutions when large fractions of clients have no labels?
>
> Please see our response to Reviewer qh3s: the unlabeled clients provide *regularization* that prevents overfitting to the labeled training clients. Degenerate solutions are prevented by a suitable choice of the regularization constant $\lambda$ in (4), as is necessary for all regularized systems.
>
> > How does the method perform when the proportion of unlabeled clients increases significantly (e.g., 70–90%)? Additional experiments varying this ratio would help understand the robustness of the approach.
>
> FlowDUP is very robust to this, please see our main results in Table 1, 2, 3 and 4, where we already reports results with 80% and 90% unlabeled clients (columns $p=0.2$ and $p=0.1$, respectively), as well our ablation studies in the appendix.
>
> > How does the proposed method scale with a large number of clients and higher-dimensional models? Are there communication or optimization challenges in such scenarios?
>
> FlowDUP has no scaling issues with respect to the number of clients, as (in contrast to other works), it does not keep any client-specific models or descriptors on the server. The communication cost depend on the number of clients chosen in each update step (line 4 in Algorithm 2), not on their total number. Regarding convergence, FlowDUP is an instance of FedAvg, so we expect more clients to actually improve convergence.
>
> > Could the authors compare their approach with more recent parameter-efficient personalization methods in federated learning?
>
> Please see Section 6 and Appendix C for a discussion of how FlowDUP relates to prior pFL work. Experimentally, we already compare FlowDUP with all prior methods that can create personalized methods for clients with only unlabeled data.
>
> > Is there any interpretation or analysis of the learned low-dimensional parameter space? For example, does it capture meaningful client variation?
>
> This is a misunderstanding, see above. The parameter space is fixed based on a random matrix, it is not learned.

---

> > ### Author Rebuttal · Reviewer_Xaig · 2026-04-02
> >
> > Thank you for the rebuttal. You have addressed most of my concerns, particularly clarifying the role of unlabeled data (regularization to prevent overfitting), the robustness to high proportions of unlabeled clients (already in Tables 1-4), and the fixed random projection (not a learned subspace).
> >
> > However, I maintain my original score of 4 (Weak Accept). The paper is technically solid and addresses an important problem, but the novelty relative to existing personalized FL literature remains limited, and the comparison with recent parameter-efficient personalization methods could be expanded. I look forward to seeing the final version with the promised clarifications (e.g., moving system metrics to the main text, additional reproducibility details).

---

### Official Review · Reviewer_qh3s · 2026-03-11

**Soundness:** 3
**Presentation:** 3
**Significance:** 4
**Originality:** 3
**Overall Recommendation:** 4
**Confidence:** 3

**Summary:**

The paper studies personalized federated learning, in a setting where only a fraction of clients have labeled data, while many clients possess only unlabeled data. The proposed method, named FLowDUP, uses a hypernetwork framework where personalized models are generated based solely on the clients unlabeled data. A PAC-Bayesian generalization bound is provided in the transductive multi-task learning setting. The bound is used to motivate the design choice of the loss function. The method is then evaluated on several heterogeneity settings and compared to other federated learning baselines. The evaluations shows higher accuracy, particularly when the fraction of labeled clients is small.

**Compliance With Llm Reviewing Policy:**

Affirmed.

**Key Questions For Authors:**

1. During training, the gradients sent by the unlabeled clients aim only to make the personalized models generated by the hypernetwork $v_i = h(X_i,\psi_h)$ closer to the regularized model $\psi_r$ (which is also learnable). Can the authors provide any intuition to why this could be helpful?
2. Does it make sense for clients with unlabeled datasets to be given the same weight in the training? Does that help the labeled clients too?
3. Can the authors provide more details about the evaluation? Does the model see the clients unlabeled data that are to be tested?
4. Can the authors address the weaknesses mentioned above?

**Limitations:**

yes

**Strengths And Weaknesses:**

## Strengths

1. The proposed method consistently achieves higher accuracy than the considered baselines across multiple datasets
2. The paper provides a PAC-Bayesian generalization analysis for the proposed framework, which motivates the structure of the model and the regularization term used during training.
3. By generating model parameters from a low-dimensional representation through a hypernetwork and random expansion, the method reduces the effective number of learned parameters.
4. The method generalizes the hypernetwork framework to generate personalized models based only on unlabeled data.

## Weaknesses
1. The theoretical analysis only shows the generalization bounds and does not establish convergence guarantees. Furthermore, it is not clear under which assumptions the convergence of the method would hold and whether the performance gain can be theoretically shown in the presence of heterogeneity.
2. The comparison with personalized federated learning methods is limited. The experimental results almost seem too good to be true, with suspiciously high performance even in the setting without unlabeled data ($p=1$). This might suggest that the setting is not favorable to single global model methods, and makes it difficult to determine to what extent the improvements stem from leveraging unlabeled clients versus the architectural advantages of the hypernetwork-based personalization.
3. The code was not provided and some reproducibility details are lacking (e.g. the choice of the GradientUpdate function)

---

> ### Author Rebuttal · Authors · 2026-03-30
>
> Thank you for your review and the positive assessment of our work. We hope you will continue to support our contribution during the upcoming discussion and decision phase.
>
> > The theoretical analysis only shows the generalization bounds and does not establish convergence guarantees. Furthermore, it is not clear under which assumptions the convergence of the method would hold and whether the performance gain can be theoretically shown in the presence of heterogeneity.
>
> The convergence properties are standard because the optimization is an instance of FedAvg, see our response to Reviewer qjJJ. We’ll clarify this and state them explicitly in the manuscript.
>
> > The comparison with personalized federated learning methods is limited.
>
> We are studying the setting in which future clients might have no labeled data, and compare to all prior methods that we are aware of that can handle this situation. Of course, there are many more pFL methods, but these are not applicable in our problem setting.
>
> > The experimental results almost seem too good to be true, with suspiciously high performance even in the setting without unlabeled data (p=1).
>
> This seems to be a misunderstanding: $p=1$ means that all training clients have labeled data. Therefore, it is the setup with most information available at training time (compared to $p<1$), so it is not surprising that the results are the best for it (across all methods).
>
> > This might suggest that the setting is not favorable to single global model methods, and makes it difficult to determine to what extent the improvements stem from leveraging unlabeled clients versus the architectural advantages of the hypernetwork-based personalization.
>
> The question of how well single global models work in the individual setting is exactly why we also report results for FedAvg under the same conditions. Indeed, FedAvg generally performs reasonably but not very well in the settings we study. This is unsurprising, though, as otherwise personalization would simply not be required.
>
> For results on how much unlabeled clients help, please see our ablation studies in Tables 7, 8 and 9 in the appendix, which report consistent improvements.
>
>
> > The code was not provided and some reproducibility details are lacking (e.g. the choice of the GradientUpdate function).
>
> We will be happy to add more details to the appendix. The code is also available in this anonymous repository: https://anonymous.4open.science/r/FlowDUP-A2DE.
>
> Specifically, in our experiments the GradientUpdate was simply pytorch’s Adam update.
>
>
> > During training, the gradients sent by the unlabeled clients aim only to make the personalized models generated by the hypernetwork $v_i = h(X_i,\psi_h)$ closer to the regularized model $\psi_r$ (which is also learnable). Can the authors provide any intuition to why this could be helpful?
>
> These gradients provide a *regularizing* effect that prevents the hypernetwork from *overfitting* to the training clients. Intuitively, pushing the produced models towards a shared base model means that the hypernetwork cannot simply memorize good models for the training clients, but it  must produce models that 1) work well for the clients with labels, and 2) look similar to each other, and therefore can be expected to generalize to future clients. The first aspect can only be enforced on labeled clients, but for the second, unlabeled clients can also be exploited.
> The theory section and the PAC-Bayesian bound formalize this intuition.
>
> > Does it make sense for clients with unlabeled datasets to be given the same weight in the training? Does that help the labeled clients too?
>
> Please, note that using sampling parameter $\alpha$ we can implicitly balance how much unlabeled clients should affect training. Also, we would expect that putting more weight on training tasks with labels could lead to improved models specifically for these, but since our goal is to produce personalized models for (test-time) clients that might not have labels, we did not explore this option.
>
> > Can the authors provide more details about the evaluation? Does the model see the clients unlabeled data that are to be tested?
>
> Thanks for bringing this up, this is an important aspect that we want to make sure is not misunderstood. We use a three-way split of the data to avoid any risk of cross-contamination.
>
> 1- during training, the clients that will be used for testing are not used in any form
>
> 2- at prediction time, for each such test client, we pass one batch of its unlabeled data  through the hypernetwork to produce a personalized model.
>
> 3- the quality of the resulting model is measured on the remaining client data (note that this step uses test client data labels, but only after the model has been created for the purpose of being able to report its quality).
>
> Will will make this more explicit in the manuscript.

---

> > ### Author Rebuttal · Reviewer_qh3s · 2026-04-03
> >
> > Thank you for the rebuttal response that clarifies most of my concerns. I maintain my already positive score.

---

### Official Review · Reviewer_qjJJ · 2026-03-12

**Soundness:** 3
**Presentation:** 3
**Significance:** 2
**Originality:** 2
**Overall Recommendation:** 4
**Confidence:** 3

**Summary:**

This paper studies how to generate personalized models for clients that possess only unlabeled data in federated learning. The proposed method, FLowDUP, combines a hypernetwork with a low-dimensional subspace parameterization to produce client-specific models from unlabeled data in a single forward pass. The authors provide a theoretical generalization bound in a transductive multi-task learning framework, which motivates their training objective. The empirical evaluation covers multiple datasets with different types of heterogeneity and varying fractions of labeled clients.

**Compliance With Llm Reviewing Policy:**

Affirmed.

**Final Justification:**

The rebuttal addressed my concerns about the paper. So I decided to increase the score to 4.

**Key Questions For Authors:**

See the Weaknesses part above. Moreover,
- In Algorithm 1, in every round, the server selects a subset of clients with a fraction $\alpha$ labeled. Should this fraction be related to the ratio of unlabeled clients and labeled clients? Furthermore, while there is a sensitivity analysis in numerical experiments on $\alpha$, there is no theoretical analysis on the purpose of this hyperparameter. How would it affect the convergence or the optimization error of the algorithm?

**Limitations:**

yes

**Strengths And Weaknesses:**

Strengths:
- The problem is well-motivated and the approach is novel.
- The paper is generally well-written and structured.

Weaknesses:
- Only a generalization bound is provided in the paper, while the optimization error is overlooked. The training procedure of the proposed method is different from previous work (it requires sampling unlabeled data during training), and hence, the optimization error bound could not be covered by previous work. It would be beneficial to analyze the optimization error bound for a deeper understanding of the algorithm.
- It is claimed that the generalization bound motivates the design of the objective function. However, there is no proof that minimizing the empirical objective tightens the bound, or that the chosen hyperparameters correspond to optimal values in the bound. Specifically, the regularization parameter $\lambda$ is tuned in the experiment rather than set from the theoretical insights.
- The numerical experiments are performed on small-scale datasets and model architectures.

---

> ### Author Rebuttal · Authors · 2026-03-30
>
> Thank you for your review, which will help us to improve our presentation. In the following, we provide answers to your questions.
>
> > Only a generalization bound is provided in the paper, while the optimization error is overlooked. The training procedure of the proposed method is different from previous work (it requires sampling unlabeled data during training), and hence, the optimization error bound could not be covered by previous work.
>
> This is a misunderstanding. The optimization procedure is an instance of standard federated averaging, the difference to prior work lies only in the objective functions. The objective based on the unlabeled data still qualifies for the guarantees of federated learning, and existing guarantees still hold. Note that in the optimization literature there is no assumption on the existence of labels. The assumptions there are on a higher level on loss functions, i.e. smoothness, etc, which does hold in our setting for the labeled and unlabeled clients objective functions. We will add a formal statement of this fact.
>
>
> > It is claimed that the generalization bound motivates the design of the objective function. However, there is no proof that minimizing the empirical objective tightens the bound, or that the chosen hyperparameters correspond to optimal values in the bound. Specifically, the regularization parameter is tuned in the experiment rather than set from the theoretical insights.
>
> We do not find this a fair criticism. We do not claim that our algorithm precisely implements the theoretical bound, we claim that the bound *motivates* the algorithm, and that it does.
> Note that our objective function has the exactly same terms as the our bound, i.e. Theorem 4.1, and generally minimizing our objective would result in minimizing the bound approximately. Regarding regularization parameter: generalization bounds are necessarily pessimistic, because they describe worst-case guarantees and they do not take computational restrictions into account. For real-world experiments it is common practice to adjust constant to reflect that real-world data is not worst-case, and to make approximations that ensure computational tractability.
>
>
> > The numerical experiments are performed on small-scale datasets and model architectures.
>
> We follow standard setups from the literature, in particular prior works (Scott et.al., 2024) and (Ye et. al, 2024). Federated learning aims at training efficient models for clients that have limited resources and therefore need to cooperate to achieve high utility models. In fact, our setup is already more heterogeneous in type and scale than most studied: we report results on class heterogeneity as well as feature heterogeneity for up to 3500 clients with 800.000 datapoints. We report two common network architectures, and various subspace dimensions. Of course, even more experiments would always be beneficial, but we believe the current evaluation is already meaningfully broad.
>
> > In Algorithm 1, in every round, the server selects a subset of clients with a fraction $\alpha$  labeled. Should this fraction be related to the ratio of unlabeled clients and labeled clients? Furthermore, while there is a sensitivity analysis in numerical experiments on $\alpha$, there is no theoretical analysis on the purpose of this hyperparameter. How would it affect the convergence or the optimization error of the algorithm?
>
> In our algorithm, the labelled clients contribute to both supervised loss and regularization, while unlabelled clients only contribute to regularization. The hyperparameter $\alpha$ balances these, making sure that the supervised signal is used properly, while not overfitting to labelled clients. A very low value of $\alpha$ could hurt the optimization, however based on our ablations (Figure 10.) the algorithm (including the optimization) is robust to this hyperparameter and no extensive tuning is needed.

---

> > ### Author Rebuttal · Reviewer_qjJJ · 2026-04-03
> >
> > Thank you for your response. It addressed my concerns, particularly regarding the optimization error and the ratio of unlabeled clients. I will update my score accordingly.

---

### Decision · Program_Chairs · 2026-04-30

**Decision:**

Accept (regular)

**Comment:**

This paper investigates the problem of personalized federated learning for scenarios where participating clients only possess unlabeled data. To address this, the authors propose a method that generates client-specific models using a hypernetwork that outputs a low-dimensional subspace parameterization, relying only on a forward pass of the unlabeled data. The objective function of this method is theoretically motivated by a transductive multi-task PAC-Bayesian generalization bound.

Despite the positive feedback, the reviewers pointed out a few limitations. Specifically, the algorithmic novelty is somewhat limited relative to existing parameter-efficient and hypernetwork-based personalization methods in the literature.

Furthermore, the initial submission lacked explicit discussion on optimization convergence guarantees and concrete system metrics like communication payload, though the authors clarified these aspects to some extent during the rebuttal phase.

Furthermore, although the method is grounded theoretically with a PAC-Bayesian bound tailored for mixed labeled and unlabeled clients, the theory heavily relies on the results of Zakerinia et al. with limited novelty.

Despite these limitations, there is a unanimous consensus among the reviewers that the merits of the paper outweigh its weaknesses.

The paper addresses a realistic scenario in FL where label acquisition is costly or impossible.

Also, the combination of hypernetworks with low-dimensional random projections leads to good empirical performance across several statistically heterogeneous settings.